# Comparison of the human microbiome in adults and children with chronic rhinosinusitis

**Il-Ho Park[1,2]ᵒ, Joong Seob Lee[3]ᵒ, Joo-Hoo Park [1,2], Sung Hun Kang[4], Seok Min Hong[5], Il Seok Park[5], Joo Heung Yoon[6], Seok Jin Hong [5]\***

**1** Department of Otorhinolaryngology-Head and Neck Surgery, Guro Hospital, Korea University College of Medicine, Seoul, South Korea, **2** Upper Airway Chronic Inflammatory Disease Laboratory, Korea University College of Medicine, Seoul, South Korea, **3** Department of Otorhinolaryngology-Head and Neck Surgery, Hallym Sacred Heart Hospital, Hallym University College of Medicine, Anyang-Si, Korea, **4** Department of Biomedical Sciences, Hallym University College of Medicine, Chuncheon, Korea, **5** Department of Otorhinolaryngology-Head and Neck Surgery, Dongtan Sacred Heart Hospital, Hallym University College of Medicine, Hwaseong-Si, Korea, **6** Division of Pulmonary, Allergy, and Critical Care Medicine, Department of Medicine, University of Pittsburgh Medical Center, Pittsburgh, PA, United States of America

ᵒ These authors contributed equally to this work.
\* enthsj@hanmail.net

**Data Availability Statement:** Data are uploaded in NCBI Sequence Read Archive, SRA and the accession number is SUB 8273275. (BioProject ID: PRJNA668045).

## Abstract

We hypothesized that differences in the microbiome could be a cause of the substantial differences in the symptoms of and treatment options for adult and pediatric patients with chronic rhinosinusitis (CRS). First, we characterized the differences in the nasal microbiomes of pediatric and adult CRS patients. Swabs were obtained from 19 patients with chronic rhinosinusitis (9 children and 10 adults). The bacterial 16S rRNA gene was pyrosequenced to compare the microbiota of the middle meatus. No significant differences were found in species richness and alpha-diversity indices between the two groups. However, in the comparison of diversity between groups using the unweighted pair group method with arithmetic mean (UPGMA) clustering of microbiome taxonomic profiles, we observed a relatively clear separation between the adult and pediatric groups. Actinobacteria had a significantly higher relative abundance in the adult group than in the pediatric group at the phylum level. At the genus level, Corynebacterium showed significantly higher relative abundance in the adult group than in the pediatric group. This is a comparative study between the microbiomes of adult and pediatric CRS patients. We expect this study to be the first step in understanding the pathogenesis of CRS in adults and children using microbiome analysis.

## Introduction

Chronic rhinosinusitis (CRS) is a chronic inflammation of the nasal and paranasal sinuses, persists for more than 12 weeks, and is accompanied by symptoms such as nasal obstruction, congestion, discharge, cough, and loss of smell [1]. The prevalence of adult CRS in the Korean population was 8.4% in a study analyzing 5-year cross-sectional data from the Korean National Health and Nutrition Examination Survey [2]. Results from the National Health Interview Survey of the United States (US) also reported similar results (12.5% of the US population) [3]. Due to the high prevalence and close connection between its symptoms and daily life, CRS

**Funding:** The following institutions provided funding for the study in the form of grants: Korea University Guro Hospital "KOREA RESEARCH-DRIVEN HOSPITALS", grant O1905541, awarded to IHP; Young Researcher Program through the National Research Foundation of Korea (NRF) funded by the Ministry of Science and ICT (MSIT), grant NRF-2018R1C1B6008596), awarded to SJH.

**Competing interests:** The authors have declared that no competing interests exist.

accounts for substantial health care expenditures in terms of office visits, antibiotic prescriptions filled, lost work days, and missed school days. Treatment strategies for CRS patients are limited due to their heterogeneous pathology. Disturbance of the nasal microbiome is proposed as a new strategy to overcome CRS [4].

The microbiota, the microorganisms that live inside and on humans, contain nearly ten times more cells than human somatic and germ cells combined. Recently, newly developed tools, such as high-throughput sequencing, have allowed us to begin to appreciate the role of the microbiota by investigating the members of a microbial community [5, 6]. Now, we understand that dysfunctions of the human microbiota are linked to various diseases, and the potential of the human microbiome as an early detection biomarker and target for therapeutic intervention is a vibrant area of current research [7]. The microbiome is also attracting attention with regard to its role in the development or progression of disease in the study of CRS pathogenesis [8].

Pediatric CRS differs significantly from adult CRS in terms of clinical features [9]. The symptoms of pediatric CRS differ from those of the adult CRS. According to the 2012 European position paper on rhinosinusitis and nasal polyps (EPOS) guidelines, persistent cough is an important symptom of CRS in children. In addition, unlike that in the adult CRS group, the mainstay of therapy in the pediatric CRS group is medical treatment and surgery is reserved for a relatively small number of patients who do not respond to medical treatment [10]. It is generally believed that the difference is due to anatomy, histopathology, state of the immune system, and effect of certain predisposing factors/comorbidities, such as frequent viral upper respiratory tract infections, and enlarged adenoidal pads [11, 12]. We hypothesized that differences in the microbiome could be a cause of the substantial differences between the adult and pediatric populations. In particular, we assumed that the nasal microbiome differed between the pediatric and adult patients with CRS. Therefore, in the present study, we investigated the bacterial abundance and diversity in children and adults with CRS and evaluated the differences between the two groups.

## Materials and methods

### Sample collection

Intraoperative swabs were obtained from 19 patients (9 children and 10 adults) and taken after general anesthesia, from the middle meatus and/or anterior ethmoid region in all patients. Samples were collected carefully to avoid contamination from the anterior nostril, nasal vestibule and nasal cavity. During swabbing, we used nasal speculum and endoscope and did not touch the anterior nostril and nasal vestibule. Samples were immediately placed on ice and frozen at − -80˚C. The diagnosis of CRS was based on historical, endoscopic, and radiographic criteria and CT findings of sinuses according to the 2012 European position paper on rhinosinusitis and nasal polyps (EPOS) guidelines. The symptoms of rhinosinusitis in adults included nasal obstruction and discharge (ant. or post.), facial pain/pressure, and reduction of sense of smell. However, the symptoms of rhinosinusitis in children included nasal obstruction and discharge (ant. or post.), facial pain/pressure, and cough. CRS diagnosis was confirmed when the above-mentioned symptoms lasted for more than three months. None of the patients had taken oral steroids, non-steroidal anti-inflammatory drugs, antihistamines, or antibiotics for at least four weeks. Patients who smoked were excluded. Patients with pregnancy, immunocompromised, trauma, previous head and neck radiation, and other sinonasal diseases, such as acute rhinosinusitis, fungal sinusitis, and tumors were excluded. Pediatric patients with primary ciliary dyskinesia or cystic fibrosis were also excluded from the study. All patients were recruited from the Department of Otorhinolaryngology, Hallym University Dongtan Sacred

Heart Hospital, Korea. Informed consent was obtained according to the Declaration of Helsinki. This study was approved by the Hallym University Institutional Review Board, which also authorized the research, and was carried out in accordance with the guidelines of the Human Ethics Committee of Hallym University Dongtan Sacred Heart Hospital. (2016-524-I)

### DNA extraction and pyrosequencing of the 16S rRNA gene

DNA was extracted using a DNeasy Blood & Tissue Kit (Qiagen, Hilden, Germany). The entire contents of the swab tube were carefully poured into a 1 mL sterile tube. Total DNA of all collected samples was extracted using enzymatic and mechanical protocols. DNA concentration and purity were measured using a UV-VIS spectrophotometer (Quawell, CA, USA). Extracted DNA was stored at -70°C until sequencing. DNA samples from the pediatric and adult groups were subjected to pyrosequencing. Polymerase chain reaction (PCR) amplification was performed on extracted DNA using primers targeting the V3 to V4 regions of the 16S rRNA gene. For bacterial amplification, primers 341F (5′-TCGTCGGCAGCGTC-AGATGTGTATAAGAG ACAG-CCTACGGGNGGCWGCAG-3′) and 805R (5′-GTCTCGTGGGCTCGG-AGATGTGTAT AAGAGACAG-GACTACHVGGGTATCTAATCC-3′) were used. Amplifications were carried out under the following conditions: initial denaturation at 95°C for 3 min, followed by 25 cycles of denaturation at 95°C for 30 seconds, primer annealing at 55°C for 30 s, and extension at 72°C for 30 s, with a final elongation at 72°C for 5 min.

Then, a secondary amplification was performed with the i5 forward primer (5′-AATGATA CGGCGACCACCGAGATCTACAC-XXXXXXXX-TCGTCGGCAGCGTC-3′; X indicates the barcode region) and i7 reverse primer (5′-CAAGCAGAAGACGGCATACGAGAT-XXXXXXXX-A GTCTCGTGGGCTCGG-3′) to attach the Illumina NexTera barcode. The conditions of the secondary amplification were identical to the former ones, except that the amplification cycle was set to 8. The PCR product was confirmed by 2% agarose gel electrophoresis and visualized under a Gel Doc system (BioRad, Hercules, CA, USA). The amplified products were purified using the QIAquick PCR Purification Kit (Qiagen, Valencia, CA). Equal concentrations of purified products were pooled and short fragments (non-target products) were removed using an Ampure beads kit (Agencourt Bioscience, Beverly, MA). The quality and product size were assessed on a Bioanalyzer 2100 (Agilent, Palo Alto, CA) using a DNA 7500 chip. Mixed amplicons were pooled and sequenced at Chunlab, Inc. (Seoul, Korea) on an Illumina MiSeq Sequencing system (Illumina, San Diego, CA) according to the manufacturer's instructions.

### Pyrosequencing data analysis

Processing raw reads started with a quality check and filtering of low quality (average score < 25) reads by trimmomatic 0.32. After the quality check, paired-end sequence data were merged together using PandaSeq. Primers were then trimmed with ChunLab's in-house program at a similarity cutoff of 0.8. Sequences were denoised using Mothur's pre-clustering program, which merges sequences and extracts unique sequences, allowing up to 2 differences between sequences. The Ezbiocloud database (http://www.ezbiocloud) was used for taxonomic assignment using BLAST 2.2.22 and pairwise alignment, which was used to calculate similarity [13]. Uchime and the non-chimeric 16S rRNA database from Ezbiocloud were used to detect chimeras on reads that contained less than a 97% best hit similarity rate. Sequence data were then clustered using CD-Hit7 and UCLUST8, and an alpha diversity analysis was conducted.

### Statistical analysis

Statistical analysis was performed using R version 3.1.2 (http://www.r-project.org/). The Wilcoxon rank-sum test was performed at each level (phylum, genus, and species) to confirm the

differences in the microbiomes between the two groups. The Kruskal-Wallis rank sum test and Tukey's post hoc test were used to analyze the differences in the composition of the microbiome between subgroups. Results with *a value of p* < 0.05 were considered statistically significant.

## Results

### Subjects and sequence reads counts

Swabs were obtained from 19 patients (9 children and 10 adults); pediatric patients comprised five males and four females (mean age 9.7 ± 3.7), while adult patients comprised five males and five females (mean age 46 ± 14.6). After the data was prefiltered and passed the quality check, the number of total reads and total valid reads were counted (Table 1). An average of 76,459 bacterial 16S rRNA-encoding gene sequence reads from pediatric CRS patients and an average of 70,606 from adult CRS patients were obtained. Rates of valid reads out of total reads ranged from 71.7% to 99.6% in pediatric patients and from 90.4% to 98.9% in adult patients. The mean sequence length after sequence processing per sample from all patient sinus mucosa ranged from 412 to 427 bases. There was no statistical significance in the number of total reads after pre-filtering and the rates of valid reads out of total reads between the pediatric and adult groups (Fig 1(A)). These findings indicate that both patient groups had similar bacterial loads.

### Taxonomic assignments

Valid reads were assigned against reference databases at the species level. A read was defined as being successfully identified when it matched the reference database at the species level with

**Table 1. Results from sequence read counts.**

| Pediatric Patients | Read counts | | | Read lengths | | | Taxonomic assignment | |
|---|---|---|---|---|---|---|---|---|
| | Total reads | Valid reads | percentage | Min | Average | Max | No. of reads identified (Species level) | No of species found |
| HS10 | 91073 | 90590 | 99.5 | 309 | 426.8 | 461 | 87924 | 30 |
| HS42 | 144783 | 138855 | 95.9 | 302 | 416.6 | 482 | 128902 | 104 |
| HS45 | 157693 | 155148 | 98.4 | 374 | 425.4 | 456 | 145397 | 81 |
| HS48 | 33965 | 24354 | 71.7 | 315 | 413.2 | 454 | 22259 | 356 |
| HS50 | 32237 | 24838 | 77 | 380 | 413.2 | 450 | 23191 | 253 |
| HS51 | 31378 | 25629 | 81.7 | 335 | 414.7 | 449 | 23996 | 350 |
| HS57 | 48254 | 45967 | 95.3 | 372 | 416.2 | 451 | 44655 | 147 |
| HS37 | 52303 | 52089 | 996 | 303 | 426.8 | 477 | 50720 | 105 |
| HS59 | 96444 | 95590 | 99.1 | 306 | 425.2 | 473 | 92113 | 121 |
| **Adults patients** | **Read counts** | | | **Read lengths** | | | **Taxonomic assignment** | |
| | Total reads | Valid reads | percentage | Min | Average | Max | No. of reads identified (Species level) | No of species found |
| HS14 | 102470 | 96852 | 94.5 | 315 | 412.8 | 462 | 91386 | 476 |
| HS9 | 16270 | 16049 | 98.6 | 300 | 420.5 | 480 | 15272 | 43 |
| HS47 | 159721 | 158005 | 98.9 | 317 | 413.7 | 462 | 146741 | 105 |
| HS52 | 28925 | 24730 | 85.5 | 317 | 419.9 | 462 | 23529 | 175 |
| HS55 | 69432 | 62792 | 90.4 | 397 | 416.6 | 449 | 60873 | 198 |
| HS35 | 96253 | 95134 | 98.8 | 303 | 421.1 | 479 | 91303 | 211 |
| HS61 | 55479 | 54320 | 97.9 | 325 | 414.1 | 472 | 52621 | 167 |
| HS67 | 49856 | 48906 | 98.1 | 300 | 412.1 | 460 | 47004 | 61 |
| HS69 | 60011 | 59272 | 98.8 | 307 | 417 | 482 | 57941 | 100 |
| HS71 | 67643 | 66430 | 98.2 | 352 | 409.4 | 452 | 64360 | 80 |

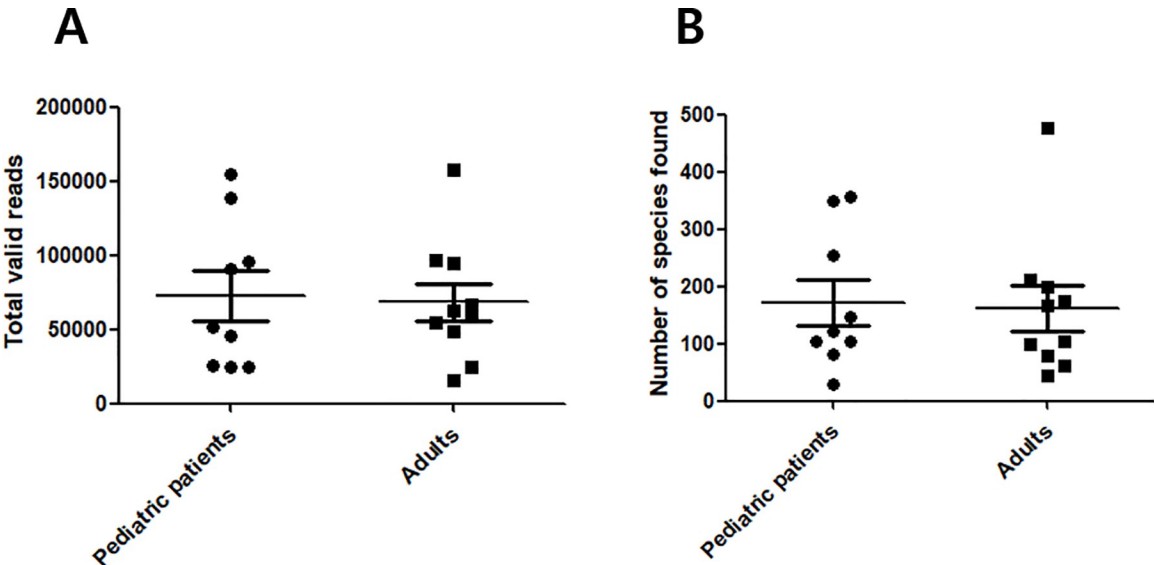

**Fig 1. Sequence read counts from middle meatal samples.** (A) Comparison of total valid reads and (B) comparison of number of species found between adult and pediatric CRS patients. There was no statistically significant difference between the two groups in both comparisons.

a 97% similarity cutoff. The number of reads identified at the species level that were obtained per sample ranged from 22,259 to 145,397 in pediatric CRS patients and from 15,272 to 146,741 in adult CRS patients. The taxonomic coverage of a database ranged from 91.4% to 97.3% in the pediatric group and from 92.9% to 97.7% in the adult group. There was no statistical significance in the number of reads identified at the species level and the taxonomic coverage between the pediatric and adult groups.

We directly counted uniquely identified species based on the reference database, and numbers from pediatric CRS patient samples ranged from 30 to 356 (mean 171.9). The number of unique species identified from adult CRS patient samples ranged from 43 to 476 (mean 161.6). There was no statistically significant difference in the number of species found between pediatric and adult groups (Fig 1(B)).

## Comparison of richness

Richness is defined as the number of unique species per sample identified using a reference database. The number of operational taxonomic units (OTUs) obtained per sample from pediatric patients in the middle meatal mucosa ranged from to 35–585 (median: 156) and 48–557 (median: 169) in the adult group. There was no statistically significant difference in the number of OTUs between the pediatric and adult groups (p = 0.744). We also checked the Chao-1 species richness indexes in intergroup comparisons, and no significant differences were found (Fig 2(A)). No significant differences were observed in other species richness indices, such as abundance-based coverage estimators (p = 0.683) and Jackknife estimation (p = 0.568) (Fig A in S1 Appendix).

Different types of statistical indices are used to describe diversity. Alpha-diversity, referring to intra-community diversity, was measured by the Shannon index, and there was no significant difference between the two groups (Fig 2(B)). Significant differences were also not observed in other alpha-diversity indices such as the NPShannon (p = 0.683), Simpton (p = 0.514), and phylogenetic diversity (p = 0.744) (Fig B in S1 Appendix). Beta diversity

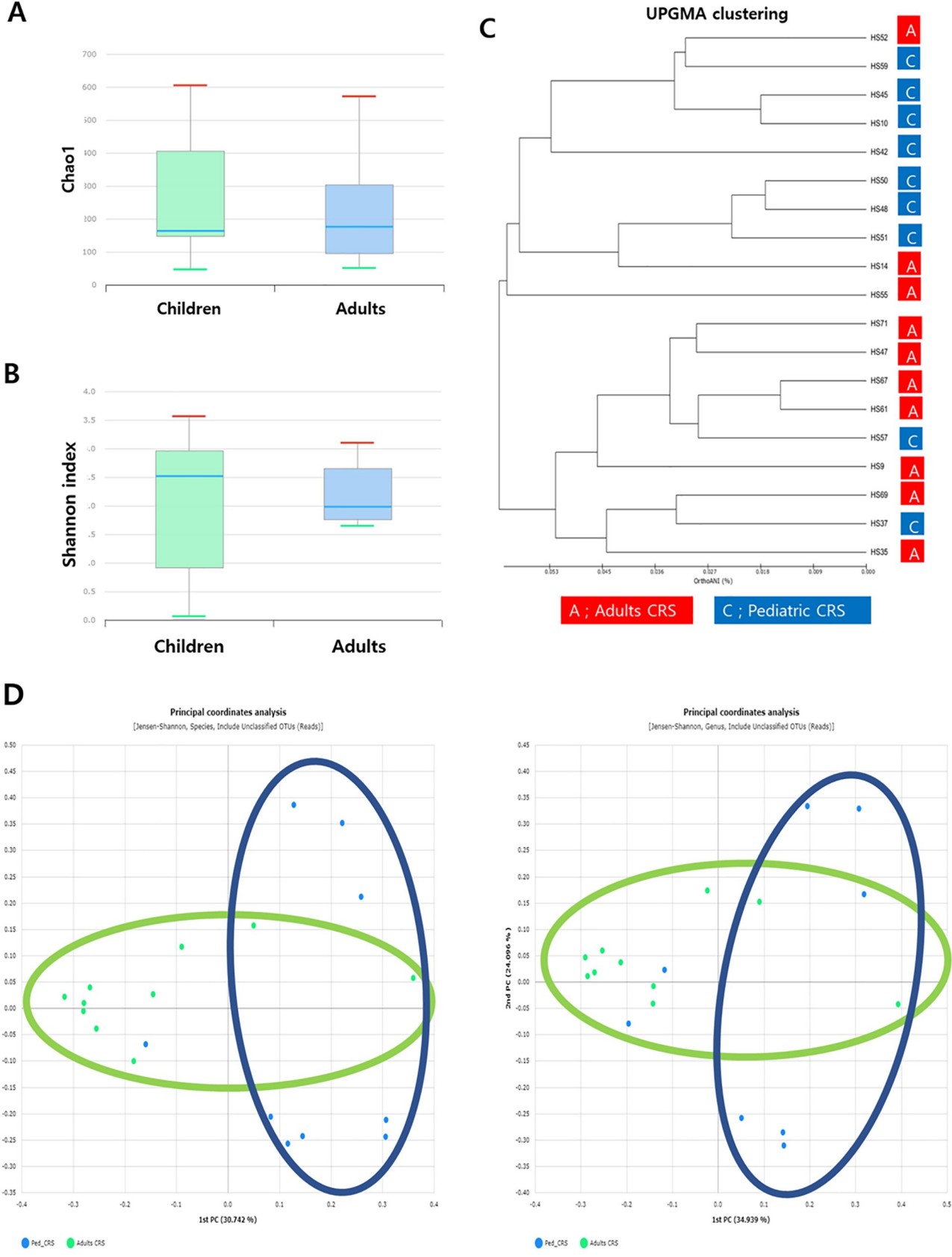

**Fig 2. A box plot of the alpha diversity indices in the adult and pediatric CRS groups.** (A) Chao1 richness values, (B) Shannon diversity indices. Overall microbial alpha diversity did not differ significantly between the two groups. (C) UPGMA clustering of microbiome taxonomic profiles in both groups using the UniFrac distance, showing a relative clear separation between the two groups. (D) Principal coordinate analysis (PCoA) plots showed two principal coordinates that explained the clear distance between samples.

showing comparison of diversity between different groups was evaluated by UPGMA clustering of microbiome taxonomic profiles in both groups using the UniFrac distance (Fig 2(C)) and principal coordinate analysis (PCoA) plots (Fig 2(D)). A relatively clear separation was observed between the adult and pediatric groups (Fig 2(C) and 2(D)).

## Composition of the CRS microbiome in pediatric and adult patients

We determined the bacterial community composition and examined differences in their relative abundance between adult and pediatric groups (Fig 3). If a species had an average relative abundance under 1%, it was classified as an "etc." At the phylum level, Firmicutes was most abundant (37.07%) in the pediatric patients, followed by Proteobacteria (33.68%), Bacteroidetes (11.00%) and Actinobacteria (9.38%). In the adult group, Actinobacteria was the most dominant bacteria and had a significantly higher relative abundance (35.96%) than in the pediatric group ($p = 0.034$). While Firmicutes (27.54%) and Proteobacteria (21.26%) were highly abundant in the pediatric group, the abundance was not significantly higher than that in the adult group (Fig 4). Streptophyta, Fusobacteria and Verrucomicrobia made up the remainder of the communities, composing a small fraction (1–6%) in the pediatric patients. Fusobacteria and Tenericutes comprise a small fraction of adult patients.

At the genus level, Haemophilus (26.8%), Staphylococcus (12.4%), Bacteroides (9.9%), and Corynebacterium (7.9%) were prevalent in the pediatric group. In the adult group, Corynebacterium was the most abundant (25.1%), followed by Staphylococcus (13.1%). On comparing the two groups, we found that only *Corynebacterium* showed a significantly higher relative abundance in the adult group than in the pediatric group.

Furthermore, at the species level, *Haemophilus influenzae* (22.0%) was the most abundant in pediatric patients. *Staphylococcus aureus* (11.9%), *Corynebacterium* group (7.6%), *Bacteroides vulgatus* (7.1%), and *Streptococcus pneumoniae* (6.8%) were prevalent in the pediatric group. In adults, *Corynebacterium spp.*, such as *Corynebacterium accolens* and *Corynebacterium tuberculostearicum* were the most abundant (23.7%), followed by *Staphylococcus aureus* (8.9%). (Fig C in S1 Appendix).

## Discussion

The association between CRS and commensal or pathogenic microbes cultured from the nasal cavity and paranasal sinuses has been investigated for a long time. Bacteriological studies in chronic rhinosinusitis are widely performed using culture techniques in the belief that certain bacteria may be pathogenic and play a role in the pathogenesis of sinusitis [14]. It is known that there can be nearly a 99% chance that the bacteria will not be cultured, and as other causes of chronic sinusitis such as mucociliary clearance, host immune response, and remodeling were highlighted more, interest in bacteriological studies has diminished [15]. Although molecular detection methods allow culture-independent investigation of microbial communities, these techniques require tremendous amounts of time and money. However, the introduction of next-generation sequencing(NGS) has not only solved most of the problems related to the previous molecular detection techniques, but it also has elicited new fields of research, including metagenomics [16, 17].

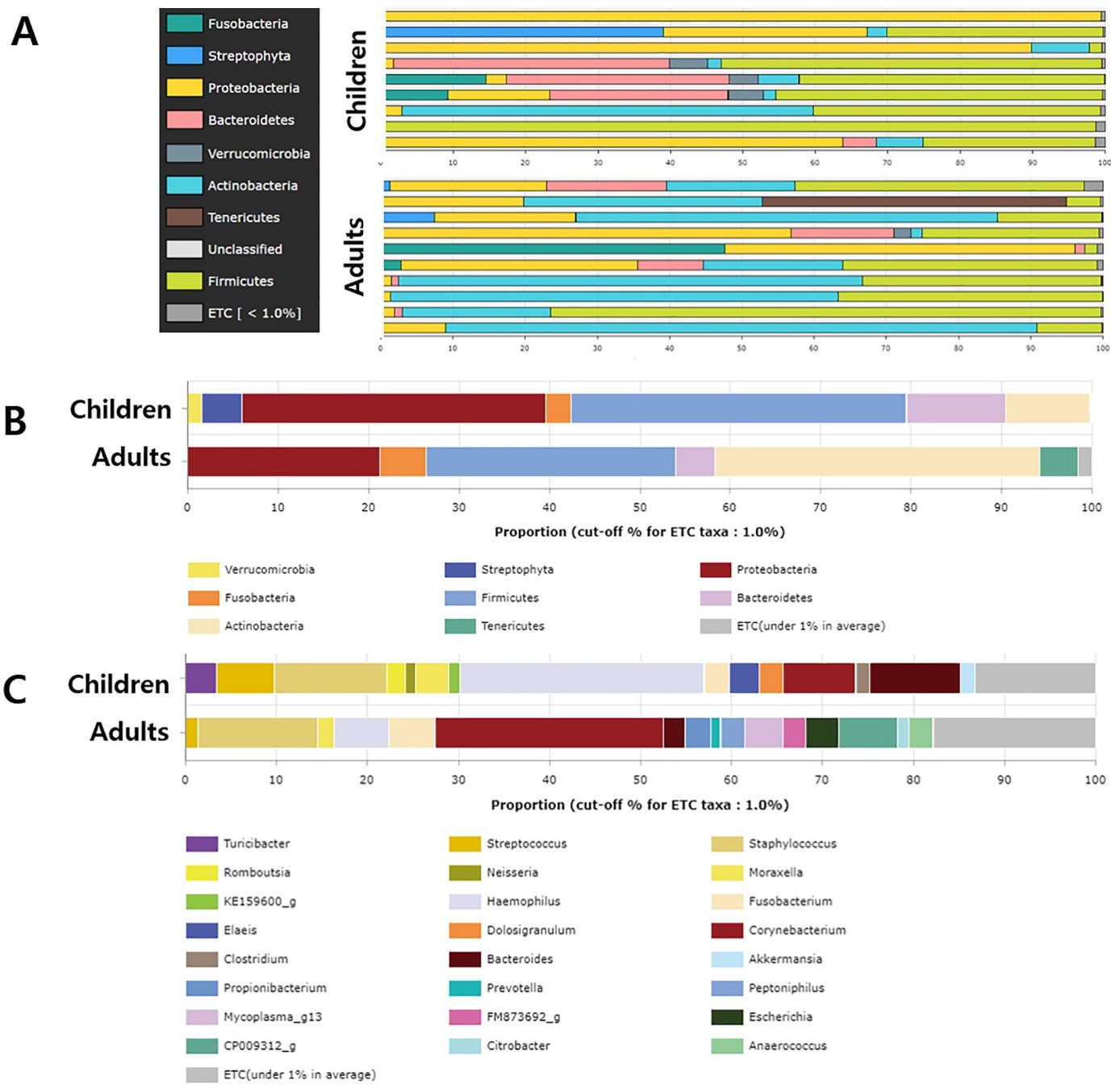

**Fig 3. Abundance of dominant bacteria in patients with CRS.** (A) Bacterial community composition at the phylum level of the nasal cavity of nine pediatric patients and ten adult patients. (B) Comparison of dominant bacterial abundance between adult and pediatric groups at the phylum level, and (C) the comparison at the genus level.

Currently, many studies that have used new NGS techniques have provided extensive evidence that the microbiome can be used to explain a substantially greater percentage of variance in the relevant phenotypes for a given condition or disease [18]. For example, the abundance of Christensenella within the human gut is negatively correlated with body mass index (BMI) and can induce weight loss when experimentally fed to mice [19]. Fecal microbiota transplant

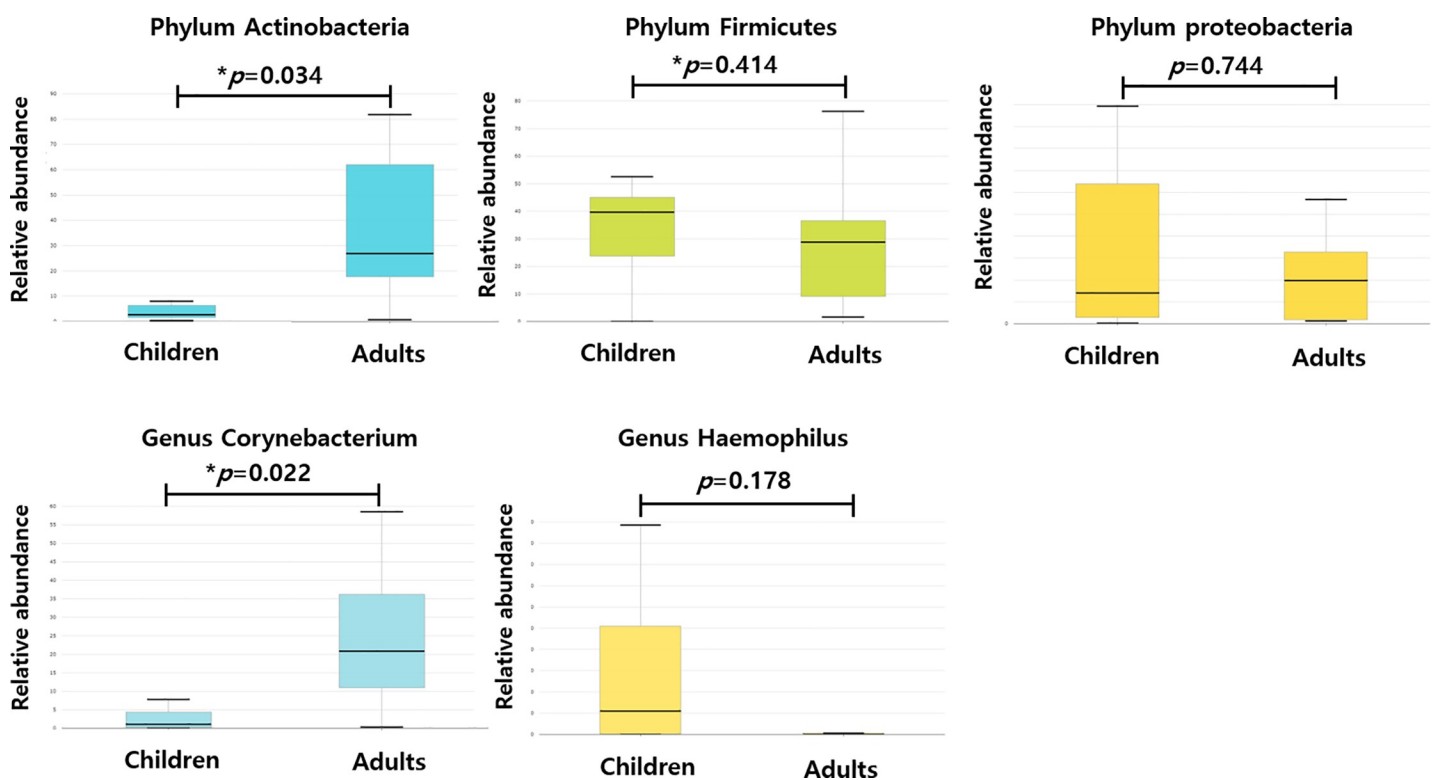

**Fig 4. Bacterial species that discriminate between adult and pediatric CRS patients.** Discriminative species that were different between the two groups were Actinobacteria at the phylum level, Corynebacterium at the genus level. *$p < 0.05$.

in humans has been associated with improvement in behavior and gastrointestinal symptoms of autism [20]. Although disease causality was not shown, as with the above diseases, an airway microbiome study has been performed and the results were viewed with great interest; chronic obstructive pulmonary disease (COPD) and asthma have been the popular target diseases for these microbiome studies. Many studies have shown associations between the respiratory microbiome and the clinical, physiological, and therapeutic features of asthma [21]. It was also shown that exacerbations of COPD are associated with changes in the respiratory microbiota and airway inflammation [22].

A recent meta-analysis of studies comparing the composition of the bacterial microbiome in adult patients with CRS showed that the most abundant bacteria across all subjects were Staphylococcus, Propionibacterium, Corynebacterium, Streptococcus, and an unclassified lineage of Actinobacteria [23]. Another systemic review of a study of adult patients' micro-biome demonstrated that despite the significant heterogeneity of studies, certain phyla including Actinobacteria, Bacteroides, and Firmicutes were consistently present [24]. Corynebacterium and Staphylococcus were the most abundant genera among all adult CRS patients (29% and 16%, respectively) in a recent study conducted in Australia [8]. Our findings were similar to those found in the literature. In the present study, the three genera that had the highest relative abundance were Corynebacterium, Staphylococcus, and Haemophilus (Table A in S1 Appendix).

The most meaningful finding in our study was that there were some differences between pediatric and adult patients in the composition of the CRS microbiome. Only one study showed that a history of acute sinusitis was associated with a significant depletion of the

nasopharyngeal microbiome in the relative abundance of taxa, including *Faecalibacterium prausnitzii* and Akkermansia spp. and enrichment with *Moraxella nonliquefaciens* [25]. We found only one microbiome study of pediatric patients with CRS in the literature; Stapleton et al. reported that *Moraxella*, *Haemophilus*, and *Streptococcus* are the most abundant taxa in pediatric patients with CRS [26]. Furthermore, there was no significant difference in the microbial composition or diversity between pediatric patients and control subjects [26]. In a study using a culture method, *Streptococcus pneumonia* (22 of 40), *Haemophilus influenzae* (14 of 40), *Staphylococcus aureus* (2 of 40), *Moraxella catarrhalis* (1 of 40), and α-hemolytic Streptococcus (1 of 40) were dominant in children with rhinosinusitis [25]. Interestingly, there was some overlap between our results and a previous study using a culture-dependent method. At the species level, our study showed that the relative abundances of *Haemophilus influenzae*, *Staphylococcus epidermidis*, *Bacteroides vulgatus*, *Corynebacterium pseudodiphtheriticum*, and *Streptococcus pneumoniae.*

In the present study, we showed that there was no statistical difference in the number of total reads, the number of reads identified at the species level and taxonomic coverage rates, and the number of species between pediatric and adult CRS patient groups. There were also no significant differences in species richness indices and alpha-diversity indexes between the two groups. However, beta diversity allowed researchers to dynamically visualize and compare two groups. In this study, UPGMA clustering of microbiome taxonomic profiles in both groups using the UniFrac distance, showed a relatively clear distinction between the two groups. All distance indices were visualized with principal coordinate analysis (PCoA) plots, which had two principal coordinates that explained the greatest distance between samples (Fig 2(D)).

At the phylum level, Firmicutes and Proteobacteria were the most abundant (>30%) in pediatric patients, and Actinobacteria, Firmicutes, and Proteobacteria were the most dominant bacteria (>20%) in the adult group. Among the dominant bacteria, Actinobacteria had a significantly higher relative abundance in the adult group than in the pediatric group. At the genus level, Haemophilus and Staphylococcus were prevalent (>10%) in the pediatric group, and Corynebacterium and Staphylococcus were dominant in the adult group. Corynebacterium had a significantly higher relative abundance in the adult group than in the pediatric group.

The limitations of our study are as follows: (1) we did not show data on healthy people in this study, (2) the sample sizes for each group were small, (3) we were unable to explain the difference between the group means, and (4) we were unable to interpret and clinically apply the antimicrobial sensitivities observed in this study. Nevertheless, this could be a significant comparative study of microbiomes between adult and pediatric CRS patients.

## Conclusions

This is a comparative study between the microbiomes of adult and pediatric CRS patients. In the genus, Haemophilus was the most common CRS microbiome in children, and Corynebacterium was the most common CRS microbiome in adults. Our results show the diversity of the human upper airway microbiome in CRS, and we expect the results of our study to help broaden the understanding of pediatric and adult CRS. Further research is needed to analyze the interactions between the human immune system and microbiota in the upper airways and CRS.

## Supporting information

**S1 Appendix This appendix contains Fig A–C.**
(PDF)

## Author Contributions

**Conceptualization:** Il-Ho Park, Joong Seob Lee, Seok Jin Hong.

**Data curation:** Joo-Hoo Park, Sung Hun Kang, Seok Min Hong.

**Formal analysis:** Il-Ho Park, Joong Seob Lee, Seok Jin Hong.

**Funding acquisition:** Il-Ho Park, Seok Jin Hong.

**Investigation:** Joo-Hoo Park, Sung Hun Kang, Il Seok Park, Joo Heung Yoon.

**Project administration:** Seok Jin Hong.

**Writing – original draft:** Il-Ho Park, Joong Seob Lee, Seok Jin Hong.

**Writing – review & editing:** Seok Min Hong, Il Seok Park, Joo Heung Yoon, Seok Jin Hong.

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
