## [Decision Letter · Decision Letter 0]

11 Aug 2020

PONE-D-20-20970

Comparison of the Human Microbiome in Adults and Children with Chronic Rhinosinusitis

PLOS ONE

Dear Dr. Hong,

Thank you for submitting your manuscript to PLOS ONE. After careful consideration, we feel that it has merit but does not fully meet PLOS ONE’s publication criteria as it currently stands. Therefore, we invite you to submit a revised version of the manuscript that addresses the points raised by both reviewers during the review process.

We look forward to receiving your revised manuscript.

Kind regards,

Alkis James Psaltis, PhD, MBBS(HONS), FRACS

Academic Editor

PLOS ONE

Journal Requirements:

2. We note that you are reporting an analysis of a microarray, next-generation sequencing, or deep sequencing data set. PLOS requires that authors comply with field-specific standards for preparation, recording, and deposition of data in repositories appropriate to their field. Please upload these data to a stable, public repository (such as ArrayExpress, Gene Expression Omnibus (GEO), DNA Data Bank of Japan (DDBJ), NCBI GenBank, NCBI Sequence Read Archive, or EMBL Nucleotide Sequence Database (ENA)). In your revised cover letter, please provide the relevant accession numbers that may be used to access these data. For a full list of recommended repositories, see http://journals.plos.org/plosone/s/data-availability#loc-omics or http://journals.plos.org/plosone/s/data-availability#loc-sequencing.

Additional Editor Comments (if provided):

Thank you for your submission on an important topic. Although studies on the sinonasal microbiome in adult patients with CRS are increasing, little work has been published on the microbiome of paediatric CRS.

Both reviewers raise relevant concerns which need to be addressed prior to the study being suitable for publication.

Reviewers' comments:

Reviewer's Responses to Questions

**Comments to the Author**

1. Is the manuscript technically sound, and do the data support the conclusions?

Reviewer #1: Yes

Reviewer #2: Yes

2. Has the statistical analysis been performed appropriately and rigorously? 

Reviewer #1: Yes

Reviewer #2: I Don't Know

3. Have the authors made all data underlying the findings in their manuscript fully available?

Reviewer #1: Yes

Reviewer #2: No

4. Is the manuscript presented in an intelligible fashion and written in standard English?

Reviewer #1: Yes

Reviewer #2: Yes

5. Review Comments to the Author

Reviewer #1: This manuscript does represent a description of a novel comparison between adult and pediatric CRS that to date has not been presented in the literature. I believe this descriptive study is a useful foundation for exploration of the differences between the two diseases at a metagenomic level. The authors' limitations are clearly stated, namely that the patient numbers are small, and it is difficult to make causative conclusions based on the data presented here. The authors make appropriate conclusions based on the data presented. There are several papers looking at acute URIs and sinusitis in children with relation to baseline microbiome composition and microbiome-based interventions, but there is little in the literature looking specifically at pediatric CRS in otherwise healthy children, so this paper is unique.

It would be additionally useful to include more discussion of what the beta diversity findings mean--ie that children's microbial communities are more similar to each other than they are to adults. This would appeal to a broader audience without as much experience in microbiome research.

There were a few changes I would make within the text:

Abstract:

"The bacterial 16S rRNA gene was pyrosequenced to compare the microbiota of the middle meatal." Would change to read middle meatus

Methods:

How was pediatric CRS defined? It would be useful to establish which symptoms/consensus statement definition was used for pediatric CRS, since these differ slightly compared to the adult European Position paper

Figure 4: Legend mentions S. epidermidis but this is not represented in the figure

Results: Page 11-12, lines 195-197

"While Corynebacterium showed a significantly higher relative abundance in the adult group, there was no difference in the abundance of Haemophilus in the adult group."

This is a little unclear. In comparison to the pediatric group?

Discussion: page 13, lines 226-235

Important to clarify that these studies were done in adult CRS patients

Discussion: page 13, lines 240-241 Similar and very recently published paper

Stapleton AL, Shaffer AD, Morris A, Li K, Fitch A, Methé BA. The microbiome of pediatric patients with chronic rhinosinusitis. InInternational Forum of Allergy & Rhinology 2020 Apr 29.

https://doi.org/10.1002/alr.22597

Throughout the discussion, there are several conclusions about the relative abundance of specific species in the adult and pediatric populations, but this data is not presented or discussed in the results. It would be useful to present this data in the results section since so much of the discussion focuses on these data

Discussion: page 14, lines 265-268

This sentence is split into two with ‘and’ between limitations 2 and 3 and a period before limitation 4. Would combine into one full sentence.

Reviewer #2: The manuscript appears to be technically sound and is well written and logical. The subject is pertinent and important in describing the microbiome in chronic rhinosinusitis (CRS) in adults and children. The methods appear to be well described. On its face, the results appear to be valid.

There are several features that bear revision.

Major Issues:

In the first sentence of the abstract, the authors state "Clinical features of pediatric CRS differ significantly from those of adult CRS", but this is not explained. Further, in the introduction on page 3 line 22, the same assertion is made. The study that is quoted (Silviu-Dan) contains a table of "Symptoms of Pediatric CRS" lists nasal blockage/obstruction/congestion; nasal discharge/cough/facial pressure, which are identical to adults. These should be explained or changed.

In the abstract, it is stated that "we hypothesized that differences in the microbiome could be a cause of substantial differences in clinical features of the adult and pediatric populations." I do not see this as an hypothesis for the manuscript. It is a subject that may be addressed in the discussion and needs to be explained as above.

There is no overt hypothesis stated. On page 4 lines 28-31, The hypothesis of differences in adults and pediatric CRS being caused by microbiome differences is made, but this is not the subject addressed in the manuscript. The authors go on to "start testing our hypothesis" by characterizing differences. "Characterizing differences" is not an hypothesis. An hypothesis should be stated.

As a limitation, the use of unsheathed swabs (even those "collected carefully to avoid contamination") should be listed. It is a weakness and calls into question the microbiologic results eg the presence of staph epidermidis which may be a contaminant from the nasal vestibule.

I personally do not believe that a conclusion should begin with a claim that was not a subject of the manuscript eg "This is the first...". The conclusion should contain information that was shown to be true in the study.

It is also my personal opinion that claims of first discovery should not be made in a scientific manuscript.

Additionally, the statement in the conclusion that "we expect the results of our study to be helpful in explaining the differences in the pathogenesis of sinusitis in children and adults" appears to be an offhand comment not elucidated in the discussion section nor substantially addressed in the manuscript.

Minor Issues

The authors use the term "nasal cavity" at times which appears to refer to the site of culture. If the site of culture is in fact the middle meatus/anterior ethmoid, this is not the nasal cavity. Terminology should be consistent and meaningful and should be standardized.

"Uncultiveability" is used on page 12 which is not a word i am familiar with. It likely refers to "uncluturability" and should be reviewed and corrected.

All data was not presented in the manuscript or in an appendix, as far as could be seen ("data not shown").

In materials and methods, subject characteristics are listed (19 patients, males/females, age, etc). This information belongs in the Results section.

6. PLOS authors have the option to publish the peer review history of their article (what does this mean?). If published, this will include your full peer review and any attached files.

Reviewer #1: No

Reviewer #2: No

---

## [Author Response · Author response to Decision Letter 0]

8 Oct 2020

PLOS One 

Comparison of the Human Microbiome in Adults and Children with Chronic Rhinosinusitis

Reviewer’s comments

Reviewer #1: This manuscript does represent a description of a novel comparison between adult and pediatric CRS that to date has not been presented in the literature. I believe this descriptive study is a useful foundation for exploration of the differences between the two diseases at a metagenomic level. The authors' limitations are clearly stated, namely that the patient numbers are small, and it is difficult to make causative conclusions based on the data presented here. The authors make appropriate conclusions based on the data presented. There are several papers looking at acute URIs and sinusitis in children with relation to baseline microbiome composition and microbiome-based interventions, but there is little in the literature looking specifically at pediatric CRS in otherwise healthy children, so this paper is unique.

Q1-1: It would be additionally useful to include more discussion of what the beta diversity findings mean--ie that children's microbial communities are more similar to each other than they are to adults. This would appeal to a broader audience without as much experience in microbiome research.

A : Thank you for your valuable suggestion. In accordance with your comment, we have modified our manuscript in discussion section and described. We also have added principal coordinate analysis (PCoA) figure.

(Page 15) However, beta diversity allowed researchers to dynamically visualize and compare two groups. In this study, UPGMA clustering of microbiome taxonomic profiles in both groups using the UniFrac distance, showed a relatively clear distinction between the two groups. All distance indices were visualized with principal coordinate analysis (PCoA) plots, which had two principal coordinates that explained the clear distance between samples (Fig 2(D)).

 Fig 2 (D). Principal coordinate analysis (PCoA) plots showed two principal coordinates that explained the clear distance between samples.

There were a few changes I would make within the text:

Q1-2: Abstract:

"The bacterial 16S rRNA gene was pyrosequenced to compare the microbiota of the middle meatal." Would change to read middle meatus

A: Thank you for your comments. We have changed the word; “middle meatal” to “middle meatus” (in the abstract).

(Page 2) The bacterial 16S rRNA gene was pyrosequenced to compare the microbiota of the middle meatus.

Q1-3: Methods:

How was pediatric CRS defined? It would be useful to establish which symptoms/consensus statement definition was used for pediatric CRS, since these differ slightly compared to the adult European Position paper

A: Thank you for indicating valuable point. We have added some sentences in the revised manuscript as follows; (in the Methods session).

(Page 3-4) The symptoms of pediatric CRS differ from those of the adult CRS. According to the 2012 European position paper on rhinosinusitis and nasal polyps (EPOS) guidelines, persistent cough is an important symptom of CRS in children.

Q1-4: Figure 4: Legend mentions S. epidermidis but this is not represented in the figure

A: Thank you for your insightful comments. Mention about S. epidermidis was inserted by mistake. Inaccurate content in the figure legend was corrected. 

Q1-5: Results: Page 11-12, lines 195-197

"While Corynebacterium showed a significantly higher relative abundance in the adult group, there was no difference in the abundance of Haemophilus in the adult group."

This is a little unclear. In comparison to the pediatric group?

A: Thanks for your valuable comments. As reviewer mentioned, above sentence is not clear. Above sentence was revised to the following

(Page 12) On comparing the two groups, we found that only Corynebacterium showed a significantly higher relative abundance in the adult group than in the pediatric group.

Q1-6: Discussion: page 13, lines 226-235

Important to clarify that these studies were done in adult CRS patients

A: Thank you for your insightful comments. We have added “adult” in the manuscript, to clarify that these studies were done in adult CRS patients.

(Page 13-14) A recent meta-analysis of studies comparing the composition of the bacterial microbiome in adult patients with CRS showed that the most abundant bacteria across all subjects were Staphylococcus, Propionibacterium, Corynebacterium, Streptococcus, and an unclassified lineage of Actinobacteria.[23] Another systemic review of a study of adult patients’ microbiome demonstrated that despite the significant heterogeneity of studies, certain phyla including Actinobacteria, Bacteroides, and Firmicutes were consistently present.[24] Corynebacterium and Staphylococcus were the most abundant genera among all adult CRS patients (29% and 16%, respectively) in a recent study conducted in Australia.[8]

Q1-7: Discussion: page 13, lines 240-241 Similar and very recently published paper. Stapleton AL, Shaffer AD, Morris A, Li K, Fitch A, Methé BA. The microbiome of pediatric patients with chronic rhinosinusitis. InInternational Forum of Allergy & Rhinology 2020 Apr 29. https://doi.org/10.1002/alr.22597

A: Thanks for your kind comments and for recommending literature. As following your comment, this paper was added to our paper in discussion section.

(Page 14) We found only one microbiome study of pediatric patients with CRS in the literature; Stapleton et al. reported that Moraxella, Haemophilus, and Streptococcus are the most abundant taxa in pediatric patients with CRS.[26] Furthermore, there was no significant difference in the microbial composition or diversity between pediatric patients and control subjects.[26]

Q1-8: Throughout the discussion, there are several conclusions about the relative abundance of specific species in the adult and pediatric populations, but this data is not presented or discussed in the results. It would be useful to present this data in the results section since so much of the discussion focuses on these data

A : Thanks for your great comments. And we have added results about the relative abundance of specific species in the adult and pediatric patients to the revised manuscript in Results section and we also have added supplement figure (Fig S1).

(Page 12) Furthermore, at the species level, Haemophilus influenzae (22.0%) was the most abundant in pediatric patients. Staphylococcus aureus (11.9%), Corynebacterium group (7.6%), Bacteroides vulgatus (7.1%), and Streptococcus pneumoniae (6.8%) were prevalent in the pediatric group. In adults, Corynebacterium spp., such as Corynebacterium accolens and Corynebacterium tuberculostearicum were the most abundant (23.7%), followed by Staphylococcus aureus (8.9%). (Fig S1)

Q1-9: Discussion: page 14, lines 265-268

This sentence is split into two with ‘and’ between limitations 2 and 3 and a period before limitation 4. Would combine into one full sentence.

A: Thank you for your great comment. We have revised the sentence and combined into one full sentence. 

(Page 15) The limitations of our study are as follows: (1) we did not show data on healthy people in this study, (2) the sample sizes for each group were small, (3) we were unable to explain the difference between the group means, and (4) we were unable to interpret and clinically apply the antimicrobial sensitivities observed in this study. Nevertheless, this could be a significant comparative study of microbiomes between adult and pediatric CRS patients.

 

Reviewer #2: The manuscript appears to be technically sound and is well written and logical. The subject is pertinent and important in describing the microbiome in chronic rhinosinusitis (CRS) in adults and children. The methods appear to be well described. On its face, the results appear to be valid.

There are several features that bear revision.

Major Issues:

Q2-1: In the first sentence of the abstract, the authors state "Clinical features of pediatric CRS differ significantly from those of adult CRS", but this is not explained. Further, in the introduction on page 3 line 22, the same assertion is made. The study that is quoted (Silviu-Dan) contains a table of "Symptoms of Pediatric CRS" lists nasal blockage/obstruction/congestion; nasal discharge/cough/facial pressure, which are identical to adults. These should be explained or changed.

A: Thank you for indicating valuable point. We thought that the clinical characteristics of CRS in the point of treatment were significantly different. For example, the mainstay of treatment in pediatric CRS is medical treatment. If it is not effective, conservative surgery, such as adenoidectomy, may be considered. But, if medical treatment fails in adults, the surgeons may consider endoscopic sinus surgery immediately. All of pediatric CRS symptoms (you have mentioned above) is correct. However, according to EPOS 2012 guideline, the smell dysfunction (not cough) is included in the adult symptoms. According to your opinion, we have modified some sentences as follows: 1) we have removed first sentence of the abstract, and added some corrections in second sentence, 2) we have added some information regarding differences in the symptoms between adult and children. 

(Page 2) We hypothesized that differences in the microbiome could be a cause of the substantial differences in the symptoms of and treatment options for adult and pediatric patients with chronic rhinosinusitis (CRS). First, we characterized the differences in the nasal microbiomes of pediatric and adult CRS patients.

Q2-2: In the abstract, it is stated that "we hypothesized that differences in the microbiome could be a cause of substantial differences in clinical features of the adult and pediatric populations." I do not see this as an hypothesis for the manuscript. It is a subject that may be addressed in the discussion and needs to be explained as above.

A: In last part of the introduction, we stated that this study is the first step toward proving our hypothesis. However, in the abstract we did not mention this. So above statement in abstract can cause misunderstanding and is not appropriate. We corrected abstract to fix it. 

(Page 2) We hypothesized that differences in the microbiome could be a cause of the substantial differences in the symptoms of and treatment options for adult and pediatric patients with chronic rhinosinusitis (CRS). First, we characterized the differences in the nasal microbiomes of pediatric and adult CRS patients.

Q2-3: There is no overt hypothesis stated. On page 4 lines 28-31, The hypothesis of differences in adults and pediatric CRS being caused by microbiome differences is made, but this is not the subject addressed in the manuscript. The authors go on to "start testing our hypothesis" by characterizing differences. "Characterizing differences" is not an hypothesis. An hypothesis should be stated.

A: Thank you for your valuable suggestion. In accordance with your comment, we have modified our manuscript in introduction.

(Page 4) We hypothesized that differences in the microbiome could be a cause of the substantial differences between the adult and pediatric populations. In particular, we assumed that the nasal microbiome differed between the pediatric and adult patients with CRS. Therefore, in the present study, we investigated the bacterial abundance and diversity in children and adults with CRS and evaluated the differences between the two groups.

Q2-4: As a limitation, the use of unsheathed swabs (even those "collected carefully to avoid contamination") should be listed. It is a weakness and calls into question the microbiologic results eg the presence of staph epidermidis which may be a contaminant from the nasal vestibule.

A: Thank you for your insightful comments. However, we thought that the risk of contamination was very low because during swabbing, we used nasal speculum and endoscope and did not touch the anterior nostril and nasal vestibule.

We have described that how we've tried to lower the risk of contamination in the Material and Methods section. 

Several researchers have reported that CRS patients had significantly increased bacterial abundance compared to control patients. S. aureus was the most prevalent organism in CRS patients, followed by S. epidermidis and Propionibacterium acnes [17]. (Mahdavinia M, Keshavarzian A, Tobin MC, Landay AL, Schleimer RP. A comprehensive review of the nasal microbiome in chronic rhinosinusitis (CRS). Clin Exp Allergy. 2016;46(1):21-41.) 

(Page 4) Samples were collected carefully to avoid contamination from the anterior nostril, nasal vestibule and nasal cavity. During swabbing, we used nasal speculum and endoscope and did not touch the anterior nostril and nasal vestibule.

Q2-5: I personally do not believe that a conclusion should begin with a claim that was not a subject of the manuscript eg "This is the first...". The conclusion should contain information that was shown to be true in the study.

It is also my personal opinion that claims of first discovery should not be made in a scientific manuscript.

A: Thank you for your insightful comments. There is only one recently published microbiome study that was done in pediatric CRS patients. (Stapleton AL, Shaffer AD, Morris A, Li K, Fitch A, Methé BA. The microbiome of pediatric patients with chronic rhinosinusitis. InInternational Forum of Allergy & Rhinology 2020, E-pub). This paper was added to our manuscript in discussion section. There are no comparative study of the microbiomes between adult and pediatric CRS patients. However, as you mentioned, the claims of first discovery should not be made in a scientific manuscript. We have modified our manuscript in discussion section.

(Page 15) Nevertheless, this could be a significant comparative study of microbiomes between adult and pediatric CRS patients.

Q2-6: Additionally, the statement in the conclusion that "we expect the results of our study to be helpful in explaining the differences in the pathogenesis of sinusitis in children and adults" appears to be an offhand comment not elucidated in the discussion section nor substantially addressed in the manuscript.

A: Thank you for your great comment. We have modified that sentence.

(Page 16) we expect the results of our study to help broaden the understanding of pediatric and adult CRS.

Minor Issues

Q2-7: The authors use the term "nasal cavity" at times which appears to refer to the site of culture. If the site of culture is in fact the middle meatus/anterior ethmoid, this is not the nasal cavity. Terminology should be consistent and meaningful and should be standardized.

A: Thank you for your comment. We have modified the terms (esp. in the figure legends.)

Q2-8: "Uncultiveability" is used on page 12 which is not a word i am familiar with. It likely refers to "uncluturability" and should be reviewed and corrected.

A: Thank you for your valuable comment. We have modified that sentence. 

(Page 13) It is known that there can be nearly a 99% chance that the bacteria will not be cultured,

Q2-9: All data was not presented in the manuscript or in an appendix, as far as could be seen ("data not shown").

A: Richness and Alpha diversity can be shown by several methods. Because all indexes showed similar results, we showed only representative indexes method in the graph. We thought graphs of other indexes are redundant, so showed only p-value of other indexes.

Q2-10: In materials and methods, subject characteristics are listed (19 patients, males/females, age, etc). This information belongs in the Results section.

A: Thank you for your valuable suggestion. In accordance with your comment, we have modified our manuscript in results section.

Results

Subjects and Sequence Reads Counts 

(Page 7-8) Swabs were obtained from 19 patients (9 children and 10 adults); pediatric patients comprised five males and four females (mean age 9.7 ± 3.7), while adult patients comprised five males and five females (mean age 46 ± 14.6).

---

## [Decision Letter · Decision Letter 1]

10 Nov 2020

Comparison of the Human Microbiome in Adults and Children with Chronic Rhinosinusitis

PONE-D-20-20970R1

Dear Dr. Hong,

We’re pleased to inform you that your manuscript has been judged scientifically suitable for publication and will be formally accepted for publication once it meets all outstanding technical requirements.

Kind regards,

Alkis James Psaltis, PhD, MBBS(HONS), FRACS

Academic Editor

PLOS ONE

Additional Editor Comments (optional):

Thank you for your revised version of your initial manuscript "Comparison of the Human Microbiome in Adults and Children with Chronic Rhinosinusitis"

I applaud your efforts for addressing the concerns of the reviewers and believe that they have all been addressed

Reviewers' comments:

Reviewer's Responses to Questions

**Comments to the Author**

1. If the authors have adequately addressed your comments raised in a previous round of review and you feel that this manuscript is now acceptable for publication, you may indicate that here to bypass the “Comments to the Author” section, enter your conflict of interest statement in the “Confidential to Editor” section, and submit your "Accept" recommendation.

Reviewer #1: All comments have been addressed

Reviewer #2: All comments have been addressed

2. Is the manuscript technically sound, and do the data support the conclusions?

Reviewer #1: Yes

Reviewer #2: Yes

3. Has the statistical analysis been performed appropriately and rigorously? 

Reviewer #1: Yes

Reviewer #2: Yes

4. Have the authors made all data underlying the findings in their manuscript fully available?

Reviewer #1: Yes

Reviewer #2: Yes

5. Is the manuscript presented in an intelligible fashion and written in standard English?

Reviewer #1: Yes

Reviewer #2: Yes

6. Review Comments to the Author

Reviewer #1: (No Response)

Reviewer #2: Comments were nicely addressed. The manuscript is technically sound, well written, and intelligible.

7. PLOS authors have the option to publish the peer review history of their article (what does this mean?). If published, this will include your full peer review and any attached files.

Reviewer #1: No

Reviewer #2: **Yes: **Andrew N. Goldberg, MD, MSCE

---

## [Editor Report · Acceptance letter]

19 Nov 2020

PONE-D-20-20970R1 

Comparison of the Human Microbiome in Adults and Children with Chronic Rhinosinusitis 

Dear Dr. Hong:

I'm pleased to inform you that your manuscript has been deemed suitable for publication in PLOS ONE. Congratulations! Your manuscript is now with our production department. 

Kind regards, 

on behalf of

Dr. Alkis James Psaltis 

Academic Editor

PLOS ONE